# Monitoring and quantifying wind turbine clutter in DWD weather radar measurements

Michael Frech<sup>1</sup>, Annette M. Boehm<sup>1</sup>, and Patrick Tracksdorf<sup>2</sup>

**Correspondence:** Michael Frech@ (Michael.Frech@dwd.de)

**Abstract.** Wind turbine clutter (WTC) is a serious threat to radar measurements from polarimetric weather radar. In Germany, wind turbines can now be built within the 5-15 km range of a weather radar in order to support and further increase the production of green energy. In order to protect the remaining 5 km radius from further wind turbine expansion, WTC is monitored and the consequences on radar data quality are quantified. There are currently no filter methods that can reliably separate wind turbine clutter from desired weather information. It is shown, that a dynamic WTC detection algorithm on the signal processor level performs well in over 80% of the time if the rotor speed of the wind turbine is larger than 5 rpm (i.e. wind turbine is running). The dynamic WTC detection algorithm fails in situations where, presumably, the static clutter contribution from the wind turbine tower is too large. This is assessed for the first time by using wind turbine operator data from two wind turbines at a distance of about 10 km from a radar system, which are provided in real-time to the Deutscher Wetterdienst (DWD). For the DWD radar system Ummendorf, persistent WTC occupies 5% of the area in the 5 km radius (approx. 4 km<sup>2</sup>), and 3% in the 5-15 km radius (approx. 18 km<sup>2</sup>). WTC is found up to an elevation of 3.5° within the 5 km radius, and 1.5° with the 5-15 km radius. Using wind measurements from a synoptic weather station near the radar system Ummendorf, a WT detection probability between 60% and 80% can be deduced for the WTs at around the 5 km radius. We show that polarimetric radar measurements are more sensitive to WTC. Especially the areal coverage of the disturbance is larger than that observed for traditional radar reflectivity. Also, the vertical extent of WTC on polarimetric moments, illustrated by the depolarisation ratio (DR), is clearly present at 3.5° elevation, and less so in radar reflectivity. A wind park at the 5 km range from the radar system Ummendorf is used to quantify the beam blockage by WTs statistically. Though the data are noisy, a 0.5 dB blockage can be estimated for this particular wind park. This is significant, as the overall accuracy of the radar reflectivity has to be within  $\pm 1$  dB, meaning 50% of the radar reflectivity error budget is consumed by this particular wind park. Further, the radar looses sensitivity in measuring precipitation, in particular at long ranges. Static beam-blockage corrections are not applicable to WTC. We conclude, that the 5 km radius must be kept free from WT expansion.

#### 1 Introduction

Radar measurements are one of the most important data sources for a wide range of applications (Fabry, 2015). They are the backbone for nowcasting products, which provide reliable warnings to the public in severe weather situations and they

<sup>&</sup>lt;sup>1</sup>Deutscher Wetterdienst, Observatorium Hohenpeißenberg, Albin-Schwaiger-Weg 10, 82383 Hohenpeißenberg, Germany

<sup>&</sup>lt;sup>2</sup>Deutscher Wetterdienst, Research and Development, Frankfurter Straße 135, 63067 Offenbach, Germany

35

55

are the only data source to provide quantitative areal information on the precipitation amount and type, which is an essential information for hydrological and flood warning applications. Radar measurements are also assimilated into numerical weather prediction models. The weather radar network operated by the Deutscher Wetterdienst (DWD) consists of 17 state-of-the-art C-band polarimetric ("dualpol") weather radar systems. Details about these systems are given in Frech et al. (2017).

Wind turbine clutter (WTC) poses a significant problem to the quality of weather radar measurements and radar based (warning) products. The World Meteorological Organisation (WMO) issued a recommendation for the protection of weather radar measurements from the influence of wind turbines (WMO, 2010). It states, among other things (Annex VI, Page 59, (WMO, 2010)): No wind turbines shall be installed within a 5 km radius of a weather radar system. At less than 5 km range, beam blockage will become significant. In the 5-20 km range, which is referred to as the "moderate impact zone", a detailed examination is recommended in order to avoid or minimise a negative effect on the measured radar data.

Since 2024, the previously restricted 5-15 km radius around radars in Germany is essentially opened for potential installations of wind turbines as part of the national transition to renewable energy. Up to 2024, the development of wind turbines in general was limited as much as possible up to a range of 15 km from the radar. This was in particular the case, when the radar beam during the operational scanning would have been, even partially, blocked by a proposed wind turbine installation. This was assessed in often time-consuming legal procedures for every proposed WT installation. Generally, it is accepted by courts, that wind turbines have a negative impact on radar data quality. But it is questioned whether warning products are affected in a such a way, that warnings become erroneous beyond an acceptable level. This refers to falsely classified warnings (see e.g. Seltmann and Böhme (2017)).

The assessment of WTC has to not only consider the effect of single wind turbines but also that of wind farms (Isom et al., 2009). The impact on radar data is not only confined to the main radar beam. Scattering of radar side-lobe emissions may affect a much larger area than just the geometric dimensions of a wind turbine. While the effect of single wind turbines may be mitigated using undisturbed surrounding radar measurements that are not affected by a wind turbine, wind farms that cover larger areas will inevitably cause data sectors without valid radar measurements. The consequence of this has to be assessed on the performance of warning algorithms. It has to be stressed that such an assessment is only valid for the particular radar warning product of interest. A generalisation of the result identified for a particular warning product to other radar based products is typically not possible.

In order to monitor and quantify the existing and future WTC situation in the vicinity of weather radars, a monitoring framework has been developed for the DWD weather radar network. An algorithm to dynamically detect WTC based on the analysis of Doppler spectra has been implemented in the radar signal processor in 2021 (Gerhards and Tracksdorf (2021)). This algorithm makes use of characteristic signatures in I&Q data and corresponding power spectra as was shown in Norin (2015).

WTC can be identified using fuzzy logic classifiers based on polarimetric moments (Ośródka and Szturc, 2022; Tang et al., 2020). The contaminated range bins are commonly thresholded. Undisturbed rangebins from the surrounding measurements of a sweep are sometimes used to fill in the resulting gaps (Tang et al., 2020) or measurements from higher undisturbed elevation are used (Ośródka and Szturc, 2022). This assumes that the precipitation field is homogeneous, which is often a fair assumption for stratiform rain events, but is not that simple for convective events. Furthermore, those approaches may work well for a small

number of wind turbines, but won't work for large wind farms with a larger number of wind turbines where the underlying assumptions (homogeneity of the precipitation field) will be violated.

A reliable filter to separate the weather signal from wind turbine clutter is not available yet. For radar reflectivity a possible approach has been presented in Norin (2017). There, a so-called natural neighbour interpolation was used to recover the weather signal in the I&Q data (Norin, 2017). An operationally usable implementation for Doppler and especially polarimetric moments is not available yet. There have been dedicated radar measurement campaigns to study typical signatures in radar moments (Lainer et al., 2021; Gabella et al., 2023). The effect of WTC on polarimetric moments and the subsequent impact on a hydrometeor classification scheme has been investigated in Frech and Seltmann (2017).

In this paper we introduce an operational wind turbine clutter detection algorithm which is operating on the signal processor level (section 2.1). We then validate this detection algorithm using wind turbine operator data from two wind turbines in about 10 km distance from a radar system (section 2.2 and section 2.3). Using the Ummendorf radar site, which has 225 wind turbines in the 15 km range, we quantify the persistence of wind turbine clutter in the operational weather radar data (section 3 and section 4). The main findings are summarised in the conclusion, where we also give an outlook on the next steps.

#### 2 Wind turbine clutter detection

# 2.1 Algorithm

90

In this section we introduce the algorithm used to dynamically detect wind turbine clutter (WTC) in the measurements from the C-band weather radar systems of DWD's weather radar network. The algorithm development was initialised and accompanied by DWD and has been presented and discussed in Gerhards and Tracksdorf (2021). Beside the main requirement to achieve a robust WTC detection up to a distance of 25 km for the operational DWD scan strategy (Seltmann et al. (2013)), there were further requirements that the algorithm could run on the hardware currently used for the signal processor (ENIGMA, details in Frech et al. (2017)) and that there would be no negative impact on the existing processing. A "simple" approach, by detecting rotor blade movement in the measurements, was considered not promising (with regard to robust operational use with a defined scan strategy) because the echoes from wind turbines fluctuate extremely (due to varying rotation speed, orientation, etc.). The approach of a more or less static WTC map was not pursued, because of the strong fluctuations of the echoes originating from wind turbines. A somehow "extended spectral clutter filter" will not be successful either, as the moving rotor-blades of a running wind turbine will, in general, add a signal on all Doppler-components, being visible as enhanced noise floor in the Doppler-spectra ("Doppler noise"). Also, a simple "polarimetric" approach, by investigating intensity differences between two receive polarizations proved not usable so far, since no assessable differences were observed (at least for the radar systems and scan strategy used by DWD). Taking into account the aforementioned requirements and "basic findings", a detailed data analysis led to an algorithm based on the real-time evaluation of I&Q-data. The algorithm was implemented as follows:

1. First, we use the discovered property that rotating wind turbines lead to an enhanced noise floor in the Doppler-spectra ("Doppler noise"). For each range gate, an estimate of the Doppler noise according to Wilfong et al. (2014) is calculated

115

and output as the derived moment "Non Coherent Power" (NCP). In the derived moment NCP, the wind turbines are already clearly visible. However, "strong fixed targets" such as towers and power lines and weather targets with a broad spectral width are also visible in the derived moment NCP and we need a robust solution to distinguish these sources.

- 2. For a "strong fixed target" a strong peak at "Doppler zero velocity" can be expected, which may be approximately 50 dB or more above the noise floor. For a wind turbine having an enhanced noise floor in the Doppler-spectra a less prominent peak at "Doppler zero velocity" is expected. To utilise this for our algorithm, a "clutter ratio" (CR) can be estimated by determining the ratio of the three central DFT components (around Doppler zero velocity) to the estimated noise. It is obvious that weather situations with a large spectral width are not addressed here.
- 3. With the knowledge that wind turbines are isolated objects, we now perform a "peak search" on the previously derived NCP for each radar ray. For the "peak search", it is most important that the NCP is available with the highest possible raw data resolution (ideally with a range oversampling, for the DWD radar systems a range (over)sampling of 25 m is available). With a pulse length of 1 μs (typical for DWD radar systems) it is expected that the wind turbine will be "visible" in approximately five successive 25 m range gates and thus will be easily detectable.
- 4. We then separate the detected peaks into "strong fixed targets" and "WTC" by applying thresholds (see table 1) to the previously estimated CR. If the estimated CR is high (>40 dB), and in addition NCP is high (> 10 dB), then the target is more likely a strong fixed target. If the measured CR is lower (

Table 1. Parameters for the algorithm used to dynamically detect WTC in the measurements from the DWD C-band weather radar systems

| parameter           | setting    | short description                                                    |
|---------------------|------------|----------------------------------------------------------------------|
| WTC_MaxRange        | 75 km      | range gates up to this radial range are inspected for WTC            |
| WTC_MaxElevation    | 5.5°       | sweeps below this antenna elevation are processed                    |
| WTC_MaxClutterRange | 25 km      | range gates up to this radial range are inspected for strong clutter |
| WTC_EstimatorSize   | 32 samples | number of samples needed to estimate Doppler-noise                   |
| WTC_CRClutter       | 40 dB      | lower limit of clutter ratio for "strong clutter"                    |
| WTC_CRMax           | 40 dB      | upper limit of clutter ratio for wind turbine                        |
| WTC_NoiseClutter    | 10 dB      | lower limit of non coherent power (NCP) for "strong clutter"         |

## 2.2 Validation

125

In Germany, new WT projects are now permitted at a distance of 5-15 km from a weather radar system to further support the increase the green energy production. If there are WTs developed within that range, developers are asked to provide the operational state and basic meteorological measurements (wind speed and wind direction, temperature) at nacelle level. In this section, we use data from two wind turbines at a distance of about 9.5 km from the radar system Türkheim (TUR) to perform an initial validation of the WTC algorithm (Figure 1).

**Figure 1.** The location of the two wind turbines (marked with TUR2 and TUR3) approximately 10 km from the radar Türkheim. Also shown are the location of other wind turbines (small dots). The 5-15 km radii are shown in 1 km steps. Map data: © GeoBasis-DE, BKG 2020 (Bundesamt für Kartographie und Geodäsie (BKG), 2020).

135

140

The nacelle is seen by the radar at an elevation of about 1°. The main beam has a width of about 150 m at this distance (with a 0.9° antenna beam width). The corresponding geometry is shown in more detail in Figure 2. The figure shows the diameter of the radar beam (here a beam width of 1° was used) at the location of the wind turbine for two different radar antenna elevation angles. The plot on the left side represents the scenario with an antenna elevation of 1.1° (terrain following precipitation scan). The plot on the right side shows the scenario with an antenna elevation of 1.5° (volume-scan). The mast of the wind turbine is 166 m high and the rotor blades have a radius of 68 m resulting in an overall height of 234 m.

**Figure 2.** Cross section of the radar-beam (red) at the wind turbine TUR2 (blue) as seen from the radar system for an antenna elevation of 1.1° (left) and 1.5° (right). Please note: The axes of the two plots are slightly different.

The WTs are in operation since May 2025. In the following analysis we consider data from 1<sup>st</sup> of June until 10<sup>th</sup> of September 2025. At least a one year data set comprising the typical meteorological situations would provide a more reliable validation. However, this shorter time period already shows valuable insights into the performance and limitations of the detection algorithm. Through the DWD data quality monitoring, we can already observe a persistent wind turbine clutter signal from these two wind turbines (Figure 3).

The following wind turbine data is available in real-time at a 10 minute resolution: wind speed (m/s) and direction (°) at nacelle height, nacelle direction (°), temperature at nacelle height (K), rotor speed (rpm) and the operational state of wind turbine (online, offline, brake, startup, shutdown). At the location of the two wind turbines, respectively, we extract the following information from the sweeps at 0.5° and 1.5° elevation and store them: the wind turbine clutter flag (1/0), CRH, CRV, NCPH, NCPV, CCORH, CCORV, TH, TV, URHOHV, UDR. This information is also stored as an average from a 3 by 3 rangebin area around each WT, but excluding the WT rangebin.

**Figure 3.** B-Plot representation of the wind turbine locations, their position relative to the antenna height using a digital elevation model (Bundesamt für Kartographie und Geodäsie (BKG), 2025), and the persistent WTC signal from the WTC monitoring. Data period shown here extends from 1<sup>st</sup> to 11<sup>th</sup> of July 2025.

We expect the detection of WTC by the detection algorithm if the wind turbine is in operation and the blades are rotating.

45 A 2-D frequency distributions of the clutter power CCOR and the clutter ration CR for the horizontal polarization H is shown in Figure 4. The clutter power is estimated by the Doppler spectrum clutter filter of the radar signal processor as the power at the 0 m/s spectral line. Large negative numbers define strong clutter power. A clutter power of -50 dB is considered to be large and is at the same time a good indicator for a very good coherency of the magnetron transmitter.

The 2-D distribution of CCORH shows a peak near -50 dB for a rotation speed smaller than 1 rpm. The WT more or less appears as a classic non-moving clutter target. Consistent with that are low values of non-coherent power (peak around 30 dB). Starting at about 5 rpm, the coherent clutter power is predominantly between -25 dB and 0 dB. This is associated with the increase of non-coherent power. The median CCORH is around -15 dB, increasing up to -10 dB (Figure 5). NCPH increases up to 60 dB for a rotation speed larger 10 rpm. The final decision whether a rangebin is classified as wind turbine clutter is based upon the CRH. The median CRH is below 40 dB with a rotation speed of 2 rpm. But the spread is large indicating the wind turbine clutter signal most likely co-exists with a strong static clutter signal from the WT tower, which is decreasing with

**Figure 4.** 2-D frequency distribution of clutter power (CCORH, upper left panel), clutter ratio (CRH, upper right panel) and non-coherent power (NCPH, lower panel center) with respect to the rotation speed of the wind turbine (10 minute average.). Radar data are from a rangebin that contains one of the WT with operation parameters are available in near-realtime. The 40 dB threshold of CRH used to classify a WT is also shown.

increasing rotation speed (Figure 6). For the data set we have analysed so far, 40% to 10% of range bins are not classified as a wind turbine clutter target depending on the rotor speed. If the rotor speed is above 6 rpm, the WT detection probability is near 90% (Figure 7). Figure 6 shows CCOR and NCPH as a function of rotor speed if there is no WTC detected by the algorithm. It corroborates the conclusion, that the static clutter level is too large so that a clear separation from WTC with this algorithm is not possible even though the non-coherent power NCPH is large. The median clutter power is near -30 dB for a rotor speed > 1 rpm. This is a strong clutter signal. The clutter threshold to separate clutter from a meteorological echo is -15 dB, so that a rangebin containing WTC and a strong static clutter signal will be detected most of the time.

Figure 5. Median, first and third quartile of CCORH, NCPH and CRH as a function of rotation speed based on the data shown in Figure 4

**Figure 6.** Median, first and third quartile of CCORH and NCPH for the cases where the WTC flag is zero (no wind turbine clutter detected) as a function of rotation speed for the data shown in Figure 4

We also show the WTC detection probability as a function of the wind speed measured at the nacelle level in Figure 7. The WTC detection probability is always lower if the wind speed is taken as a reference. Wind speed is a great proxy for the rotor speed of a wind turbine, if we can assure that the wind turbine is in operation. However, even in good wind conditions, turbines might be shut down due to maintenance, feed-in restrictions or site-specific risks to safety and wildlife. Hence, it is expected that the detection probability is lower when analysed according to wind speed than rotor speed, since it also includes the downtime of the wind turbine. This clearly highlights the benefit of wind turbine operator data for a better identification of WTC and the verification of the algorithm. The good performance of the existing WTC detection algorithm can also be deduced from Figure 3. There we show the persistent WTC detections compared to the WT locations. Most of the WT in line of sight of the radar produce a persistent WTC signal. There are no WTC detections in areas without WTs (the WTC at range

Figure 7. WTC detection probability as a function of rotor speed and wind speed measured at the nacelle level.

12 km and azimuth 60° relates to WT which are not listed in the official site data base. A satellite based product (see Figure 8) shows the presence of WT. Overall, the WTC detection algorithm obviously is not sensitive to static clutter signals.

#### 2.3 Monitoring

The wind turbine clutter detection is running in the signal processor at every DWD radar site since 2021. As soon as a sweep is acquired at the radar site, it is sent to the DWD headquarters in Offenbach and processed there. This is done for all 17 operational radar sites. First, all sweep data are quality controlled (Werner and Steinert (2012b)). Non-meteorological echoes are removed and Zh and ZDR are corrected for attenuation before radar products are computed. During quality control all detected WTC rangebins along with a selection of radar moments (see section 2.2) are stored in an InfluxDB database. This is done for sweeps of the volume scan at an elevation of 0.5, 1.5, 2.5., 3.5, 4.5. 5.5° and the terrain following precipitation scan. Extracting and storing the WTC detections allows us to monitor and quantify trends of WEA clutter in radar data primarily in a range of 15 km around a radar site.

Ultimately, this approach may provide DWD with objective arguments that clearly demonstrate the necessity to mitigate negative effects on radar data and products. It has to be shown whether compensating measurements or a regulation of WT development in the vicinity of weather radars are the right way for such mitigation efforts.

There are about 225 WT in up to range of 15 km of the radar Ummendorf. We therefore use this site to quantify the effect of WT on radar data using our WTC detection algorithm. We define WTC severity the following way: a "WTC severity" of 1 means that WTC is detected 100% of the time in the rangebin within the predefined time interval. We consider a WTC severity

195

**Figure 8.** B-plot representation of WTC severity based on WTC detections between 1<sup>st</sup> of January and 20<sup>th</sup> of January 2025 observed at an 0.5° elevation. Shown are results up to range of 15 km. Black crosses denote the location of wind turbines (229 in total) from a data based of the federal state. The circles are WT location (225 in total) based on a satellite product (Wehner et al., 2025). Only radar rangebins with WTC detected in more than 50% of time are shown (WTC severity large > 0.5). The relative height of the respective locations of WT with respect to the radar antenna height is also shown for reference using a digital elevation model (Bundesamt für Kartographie und Geodäsie (BKG), 2025)

of > 0.5 as a persistent WTC signal. It is expected that WTC severity for a given time interval will always be 

210

**Figure 9.** A comparison of the WTC severity for the lowest elevation (upper panel) and the 2.5° elevation (lower panel). The full range of severity between 0 and 1 is shown.

3.5 km). Apparently, the database of the federal state is not up-to-date for this example. There is a wind park at about 280° azimuth and in a range between 12 and 14 km, for which we do not observe a persistent WTC signal. The reason for this is that those WTs are not in line-of-sight of the radar as indicated by the orography in Figure 8.

The area occupied by WTC in the radar data within the 5 km range is about 4.5 km<sup>2</sup> or about 5% of the area. In the 5-15 km range, the total area with WTC is 18 km<sup>2</sup> (3% of the area).

Those results refer to sweeps at  $0.5^{\circ}$  elevation. The WTC signal is decreasing with elevation. Within the 5km range, WTC is seen up to an elevation of  $3.5^{\circ}$  (see Figure 9), and between 5-15 km up to an elevation  $1.5^{\circ}$ .

#### 205 3 Wind turbine impacts on radar data quality

## 3.1 Impacts at and around the WT locations

It is known, that the negative impacts on the radar measurements caused by wind turbines is not only confined to the immediate location of the WT itself. The effective volume disturbed by the WT is determined by backscatter contributions from side lobes and multi-path effects. In Figures 10 and 12 an example of a stratiform precipitation event on 30 May 2024 is shown for the radar Ummendorf. The radar sector is chosen such that it includes a wind park and an undisturbed area within the stratiform rain event. Furthermore, we have chosen an elevation of 1.5° and 3.5°. We show the uncorrected radar reflectivity factor TH (dBZ), the uncorrected depolarisation ratio UDR (dB) and the quality controlled, attenuation-corrected radar reflectivity Zh. "Uncorrected" means no clutter filter and no thresholding is applied. We chose an elevation of 3.5° because no persistent WTC signal could be found in TH for this wind park. The depolarization ratio DR (Ryzhkov et al., 2017) is a standard output from DWD's radar signal processor since 2022. The depolarization ratio is not directly measured, since we operate our radars in STAR mode, but is deduced from STAR linear polarisation measurements (Melnikov and Matrosov, 2013). In applications DR serves as a good discriminator between clutter and meteorological echoes (Kilambi et al., 2018; Michelson et al., 2020). Small

**Figure 10.** Uncorrected radar reflectivity TH in the area of a wind farm. Left panel shows data from 1.5° elevation, and the right panel data from 3.5° elevation (top row). The depolarisation ratio is shown in the bottom row. Data are taken from a stratiform rain event 30.5.2024. WT locations are denoted by crosses. All reflectivity values larger 30 dBZ are coded in red.

(large negative DR) values are expected in stratiform rain with small spherical rain drops (-20 dB or smaller). A threshold of UDR = -12 dB may be used to separate a non-meteorological echo from a meteorological echo (Kilambi et al., 2018; Michelson et al., 2020).

At  $1.5^{\circ}$  elevation, large TH values (TH > 30 dBZ) are found mainly in rangebins with wind turbines. At  $3.5^{\circ}$  elevation, the signal power and the variability over the wind turbine area is comparable to the surrounding precipitation field. For UDR, high values > -12 dB are found near wind turbines, but also in other regions. Clearly, at  $1.5^{\circ}$  elevation clutter contributions

230

**Figure 11.** Unfolded radial Doppler velocity VRADH (m/s) wind farm. Left panel shows data from 1.5° elevation, and the right panel data from 3.5° elevation (top row). The spectral width (m/s) is shown in the bottom row. See also Figure 10.

outside the wind park are present in the data. In radial direction, there is a spread of wind turbine clutter. This appears due to multi-path scattering within the wind park. Clutter-free stratiform precipitation areas (UDR values as low as -25 dB) cover a small area. At 3.5° elevation, there is no apparent influence in TH from the wind park. A clear signal is found however in UDR (between -10 and -15 dB). This shows, that polarimetric data is more affected by WTC than just the power based moments. More importantly, the resulting volume where dualpol data is affected is substantially larger. This is in line with the study of Friedrich et al. (2009). They show for different clutter types that polarimetric moments are much more sensitive to clutter. The backscatter contribution on phase and power from the side lobes is responsible for this.

240

**Figure 12.** The quality controlled radar radar reflectivity Zh in the area of a wind farm. Left panel shows data from 1.5° elevation, and the right panel data from 3.5° elevation (top row). See also Figure 10.

The radial Doppler velocity VRADH and the spectral width are shown in Figure 11. Multipath effects are visible in the Doppler velocity at 1.5° elevation, but the affected areas appear comparable to that of TH (Figure 10). At 3.5° elevation the impact of the wind farm is significantly less visible such that only a few range bins show erroneous Doppler velocities. For the spectral width (Figure 11, lower panel) the area affected by the wind farm at 1.5° elevation is larger than that of the Doppler velocity. The area affected is comparable to that of UDR. At 3.5° elevation, the wind farm has a larger effect on the spectral width. The surrounding rangebins have values on the order of 1 m/s, whereas in the presence of WTs values of 2 m/s or larger are present.

Figure 12 shows how the operational DWD radar data quality control deals with the wind park at 1.5° and 3.5° elevation. The current quality control (Werner and Steinert, 2012a), which does not make use of the WTC flag, nicely identifies WTC and thresholds the wind park area. Clutter power and the texture of differential phase already provide a clear enough signal to identify WTC. At 3.5° elevation, the operational quality control does not lead to thresholded radar rangebins. Since dualpol moments are affected at 3.5°, an impact on products based on dualpol data has to be expected.

## 4 Beam blockage due to wind turbines

The WT impact on radar moments at around the 5 km range from a weather radar has been shown in the previous section. An important result is that the dualpol data are more sensitive to the presence of WT than the radar reflectivity. In particular the vertical extent in which dualpol data are affected is significantly larger.

In 2010, the World Meteorological Organisation issued a recommendation for the protection of weather radar measurements from the influence of wind turbines (WMO, 2010) (Annex VI, Page 59). No wind turbine shall be installed within a 5 km radius. In the 5-20 km range, which is referred to as the "moderate impact zone", a detailed examination is recommended in order to avoid or minimise the negative effect on the radar data. At the 5 km range, beam blockage will become significant. Assuming a standard WT turbine geometry, the WT scattering crossection to the main beam of a radar can cause a beam blockage of 1 dB (see e.g. Argemí et al. (2012)).

Beam blockage corrections can be applied for stationary clutter targets to some extend, but provide only a constant offset correction which accounts for the loss due to beam blockage. However, any beam blockage will lead to a loss of sensitivity. This can affect the detection of (weak) precipitation at ranges further away from the radar. There are meteorological situations (e.g. freezing rain associated with small reflectivity values) where this loss of information can affect the warning process.

To quantify beam blockage from a single case study is difficult as the natural variability in the precipitation field adds too much noise to any blockage effect. Hence, we apply a statistical approach to the data of two warm seasons, analysing the reflectivity field around the wind park located south-west of the Ummendorf radar at a range of about 5 km. We expect a beam blockage effect at lower elevations for azimuths with wind turbines close to the radar, but no beam blockage effect at higher elevations and at WT-free azimuths. To test this hypothesis, we analyse two sectors, one containing WTs (180°-190°), and an adjacent sector without WTs (170°-180°), and two different elevation angles (1.5° and 3.5°) (Figure 13). The wind park sector contains 15 wind turbines. At 1.5° elevation, the main beam covers the entire turbine blade area for 6 of those 15 turbines, and only the upper part of the blade area without the nacelle for the remaining 9 turbines. At 3.5° elevation, the main beam is well above the blade area of all turbines.

We identified precipitation cases where the following conditions were met: 1) Both sectors must be fully covered with precipitation. 2) Only precipitation with mean reflectivity values between 15dBZ < ZH < 40 dBZ in the WT-free sector are considered; in doing so, drizzle and convective precipitation are excluded. No attenuation correction is applied (since we do exclude convection). 3)  $\rho_{hv} > 0.95$  in the WT-free sector. This eliminates possible clutter pixels.

For each sector and elevation angle, we compute a simple difference of the mean reflectivity values at the ranges of 6.5 km and 3.5 km, i.e. behind and ahead of the radial location of the wind park. The analysis was carried out for the two warm seasons 2021 and 2022 (1.4 - 30.9). In total 1923 sweeps met the aforementioned criteria. A histogram of the differences of the respective reflectivities (Figure 14) shows, that although the spread is quite large, the distributions appears to be normal distributions and the corresponding median reflectivity differences corroborate our expectations. At 3.5° elevation, the reflectivity difference is nearly identical (-0.32 dB). At 1.5° elevation, the difference in the sector without wind turbines is -0.45 dB, similar to the difference we find at 3.5°. In the sector with wind turbines, the difference at 1.5° is -0.95 dB which is 0.5 dB larger than the difference in the sector without wind turbines, and 0.63 dB larger than at the elevation of 3.5°. This difference is attributed to beam blockage. Data from another wind park, with only one line of wind turbines support these findings from the wind park southwest of the Ummendorf radar. This second wind park consists of one line of four wind turbines, located approximately 2.5 km west of radar Türkheim (1). The analysis was again carried out for the two warm seasons of 2021 and 2022, resulting in 1229 sweeps that met the selection criteria. Even though only one line of turbines blocks the radar beam, the beam blockage

Figure 13. Illustrating graph showing the chosen sectors to quantify beam blockage.

effect is evident in the reflectivity differences. At an elevation angle of  $3.5^{\circ}$ , the sector with wind turbines and the adjacent sector without wind turbines show differences of -0.24 dB and -0.30 dB, respectively. Whereas the reflectivity difference at  $1.5^{\circ}$  elevation in the sector with wind turbines is -0.55 dB, which is more than 0.25dB higher than at  $3.5^{\circ}$  elevation, and 0.16 dB higher than in the sector without wind turbines at  $1.5^{\circ}$ .

Weather radars are designed and operated to measure the radar reflectivity with an accuracy of 1 dB (Frech et al., 2017). It is already a challenge to achieve this on a radar system level. Additional errors due to beam blockage from WTs can lead to errors larger 1 dB which must be avoided. Increased errors will introduce additional bias to e.g. quantitative precipitation estimates. Based on the examples showed here, further installations of WT in the 5 km radius must be avoided.

#### 290 5 Conclusions

With the effort to increase Germany's production of the renewable energy, more and more wind turbines will be built within the 5-15 km radius of all weather radar of the DWD weather radar network. Wind turbine clutter poses a serious threat to the radar data quality because Doppler clutter filters do not work. This is due to the backscatter signal from the moving rotor blades of wind turbines, which spreads power across the entire velocity spectrum. As a first step to identify wind turbine disturbances in radar data, DWD has implemented a dynamic wind turbine detection algorithm in the radar signal processor. The algorithm runs operationally on every signal processor of the radar network. This detection algorithm is validated using wind turbine operator data of two wind turbines at a distance of about 10 km from the radar Türkheim. This constitutes the first operational wind turbine operator data provided to DWD. The data has been available since May 2025. A detection probability near 90% is found for wind turbine clutter if the rotation speed of the rotor blades is larger than 5 rpm. If the algorithm fails to detect

Figure 14. Statistical evaluation of beam blockage.

WTC even though the blades are rotating, it may be attributed to a strong static clutter signal from the wind turbine tower. The median clutter power is close to -30 dB which is larger than the clutter threshold of -15 dB that is used to separate a weather echo from a non-meteorological echo. Since we just recently obtained the wind turbine operator data, this result will be verified in the future for a longer period of time.

We systematically monitor the wind turbine clutter situation for every radar. We showed results from radar Ummendorf where we have numerous wind parks in the 15 km range and wind turbines as close as 3 km to the radar. Those wind parks produce a persistent wind turbine clutter signal. For some wind turbines the clutter signal is present close to a 100 % of the time. For a wind park around the 5 km range, the clutter signal is present even at 2.5° elevation.

We investigate the magnitude and area of the impact of WTC on radar reflectivity and the depolarisation ratio DR for a case study with stratiform precipitation for a specific wind park at the 5 km radius of radar Ummendorf. The WTC effect on radar reflectivity is mainly confined to the rangebins containing the WTs. In contrast, the WT disturbance of DR covers a significantly larger area. This is shown in data taken at a 1.5° elevation. Furthermore, we find a significant WT signal in DR at

 $3.5^{\circ}$  elevation. With a DR between -20 and -30 dB in the stratiform rain, an increase in DR to values between -20 and -10 dB is found. This effect is not visible in radar reflectivity. However, there might be an effect on the order of  $\pm$  1 dB which cannot be separated from the meteorological signal, considering the overall variability of Z for this case. The results suggest, that the volume impacted by WTC for dualpol moments is significantly larger than that for radar reflectivity. The effect of the wind farm on Doppler velocity for this example is comparable to that of the radar reflectivity. Spectral width however appears much more sensitive to the presence of WT. The area of impact is comparable to that of DR.

To conclude, our current, dynamic WTC detection algorithm works well for radar reflectivity, but is missing the larger impact volume for dualpol and Doppler moments. This needs to be addressed in the further development of that algorithm.

A statistical analysis was carried out to quantify the beam blockage effect caused by a wind park by using data from close to 2000 sweeps with wide-spread precipitation, excluding convective situations. The beam blockage was quantified using a reference volume close to the wind park for the same precipitation events. By analysing this large data set, a clear beam blockage signal could be found even with the underlying natural spatial variability of precipitation. We found 0.5 dB beam blockage for this specific wind park, which is considered to be significant considering the overall requirement to measure radar reflectivity with an accuracy of  $\pm 1$  dB. Because of the dynamic nature of WTC, static beam blockage corrections are not applicable. Beam blockage leads to reduced sensitivity of weather radars when it comes to measuring weak precipitation signals at far ranges (beyond 120 km). Beam blockage due to a wind park at a range of 5 km will increase with the expected repowering of WTs in the future (WT with tower heights of 300 m and "tower base diameters" of approx. 20 m are already being built).

The main conclusions and recommendations of this study are:

- The GAMIC WTC detection algorithm performs well given the variable clutter characteristics of a wind turbine
- The WTC detection algorithm needs to be adapted for polarimetric moments, because those are much more sensitive to WTC, and, in particular, a much larger atmospheric volume is impacted by WTC.
- The beam blockage results clearly suggest, that wind turbine development in the 5 km radius must be avoided.
- Wind turbine operator data are extremely helpful to improve and validate WTC detection algorithms and radar data quality control in the presence of wind turbines.

In this work we primarily focused on the detection of WTC. Correcting radar data using a filter on the IQ data level or by using compensating measurements to (partially) fill WTC contaminated radar rangebins will be further investigated in the Wivaldi Wind Turbine Clutter Experiment and Analysis (WICLEAN) project, that started in 2025. There, dedicated measurements with a X-band radar using the two wind turbines in the Wivaldi research wind park (Wildmann et al., 2022) of the German Aerospace Center (DLR, Deutsches Zentrum für Luft- und Raumfahrt e.V.) will be used to further develop mitigation approaches for polarimetric weather radars. This includes an improved detection algorithm, a filter to separate the weather signal from the WT clutter signal, and to employ additional meteorological measurements (like, among others, ombrometer

measurements and wind measurements) to add possible compensation in areas where the weather signal from radar measurements are measurements and wind measurements) to add possible compensation in areas where the weather signal from radar measurements are measurements.

Data availability. Radar data used in this work, that is not available through opendata.dwd.de can be provided upon request.

*Code and data availability.* Python code used for the analysis are available upon request. Parts of our analysis used the very useful Py-ART package (Helmus and Collis, 2016).

Author contributions. MF wrote the body of the text and carried out the analysis to validate the WT detection algorithm using WT data.

AB carried out the analysis on radar data quality and beam blockage and wrote the beam blockage section, and contributed to the text and prepared some of the figures. PT wrote the description of the WT detection algorithm and contributed to text and provided figures.

Competing interests. There are no competing interests

Acknowledgements. This work is supported by the Federal Ministry for Economic Affairs and Climate Action on the basis of a decision by the German Bundestag: project WICLEAN, Wivaldi Wind Turbine Clutter Experiment and Analysis, grant number 03EE2074A. Special thanks go to the operator of the two wind turbines near the Ummendorf radar system for providing the wind turbine operator data.

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
