# Peer review of "Monitoring and quantifying wind turbine clutter in DWD weather radar measurements"

_EGUsphere, 2025_

## Referee Comment (RC1)

This paper presented a method to detect wind turbine clutter (WTC) and validated the method through comparisons to known wind turbine locations. Moreover, the paper investigated the impact of WTC on radar data quality with a focus on polarimetric variables. The study found that polarimetric variables are more sensitive to WTC contamination and the area of impact from a wind farm is larger for the polarimetric variables compared to that for reflectivity and Doppler velocity. Additionally, the authors quantified the impact of beam blockage from wind turbines and found they induced a significant bias in reflectivity, and, as a result, strongly recommended wind farms to not be constructed within 5 km of weather radars. These findings contribute to the ongoing research on WTC mitigation and are presented clearly in the manuscript. As a result, I recommend this paper to be accepted for publication with some minor revisions.

Specific Comments:

1.  In the detection method, the three central coefficients (around zero Doppler velocity) in combination with NCP to compute CR (line 98). However, later on in the paper, a new product called clutter power (CCORH) is shown in Figs 4-6. What is the relationship between CCORH and the clutter power used in the detection method? Looking at the figures, it is easy for a reader to use CCORH as a substitute for clutter power used to determine CR, and it doesn't make sense how you can get the CR curve shown in the right panel of Fig.5 with the two curves shown in the left panel of Fig. 5.

2.  The sentence "But the spread is large indicating the wind turbine clutter signal most likely co-exists with a strong static clutter signal from the WT tower, which is decreasing with increasing rotation speed (Figure 6)" needs further clarification. I do not see a decreasing trend in CCORH in Figure 6. Also, why does the large spread in CR indicates the co-existence of WTC signal with a strong static clutter signal?

3.  For the data set shown in Fig. 4, how often was weather signals overlapping with the WTC signals? It would be interesting to break down the detection performance for cases when there is no weather versus when weather is overlapping with the WTC.

Technical Corrections:

1. Line 56, "rangebins" should be "range bins". Please check the remainder of the manuscript for similar corrections.

2. Line 96, insert comma after Doppler-spectra.

3. Line 134: I would reword the sentence as "A data set comprising of at least one year of the typical meteorological situations would provide a more reliable validation."

4. Line 142: CCORH, CCORV, TH, TV, URHOHV, and UDR needs to be defined here when they first appeared in the manuscript.

5. Line 143: "3 by 3 rangebin area" should be "3-by-3 range bin area".

6. Line 145: "ration" should be "ratio".

7. Line 156: "40% to 10%" should be "10% to 40%"

8. Line 181: "WEA" need to be defined.

9. Caption of Figure 8: "Black crosses" should be "Red crosses", "data based" should be "database", "rangebins" should be "range bins".

10. Line 191: "WEA detection algorithm" should be "WTC detection algorithm"?

11. Line 209: insert comma after "In Figures 10 and 12".

12. Caption of Figure 12: "TH" was used previously to refer to reflectivity factor in dBZ. Here "Zh" is used. Please choose one designation and stay consistent.

13. Line 267: ZH is used here for reflectivity.

14. Figure 14: "WEA" and "NoWEA" are used in the legends. I think being consistent and using "WTC" and "NoWTC" would be appropriate.

15. Line 201, add comma after "-30 dB"

---

## Referee Comment (RC3)

Figure comments  "Monitoring and quantifying wind turbine clutter in DWD weather radar measurements"
Author(s): Michael Frech et al.
MS No.: egusphere-2025-4957
2025-12-03

I am still working on writing up comments on the text, but I have some comments on figures.  Since changes to figures are more time consuming, I am going post those comments now.  See attached PDF.  I hope to get the full review up early next week (week of Dec 8).

Figure 3:  The colour bar does not correspond to the figure.  The colour bar shows pure gray colours, but the figure is using colours with a magenta/purple shade.  The figure is deceptive because starting at 0m implies the radar antenna is higher than all surrounding terrain. Google Earth says the ground height is about 800m at turbines TUR2 and TUR3, and the height at the radar is about 735m.  That means the antenna would need to be lower than the ground at that location, unless the antenna is more than 65m above ground.  (Maybe the radar is that tall?  The radar site specifications are not given.) My version of the figure using a crude topography database is below.

Figure 4  Change "wea" in title?  ("WTC"?)  Is it easy to replace the WIGOS id with a name? FYI  I could not find this ID in the OSCAR database, but I have little experience with WIGOS, so it might be somewhere else.

Figure 8:  As with Figure 3, colour bar does not correspond to the figure itself.

Figure 10  In my opinion the upper limit on reflectivity colours should be higher.  I was surprised that the turbines seemed to be spread uniformly across 3 degrees, until I realized that the observed values must be far above the colour limits; colours were saturated so no detail below 3 deg in azimuth.   I suspect the reported reflectivities are much in excess of 60 dBZ.

Figure 11  What is the meaning of white?   I can guess that clutter has exceeded the ability to correct reflectivity,  but please state meaning.   The same comment for Figure 12.

Figure 14:  What is "Wea" and "NoWea" on the figures?  Maybe "WF" (Wind Farm) was intended, since discussion in the text indicates that the distributions labelled "Wea" are from the wind farm sector.  (Elsewhere WEA is WTC, but I would prefer WF here.)

[Figure]

Coarse view of topography around TUR radar; SRTM30 data (~90m) as B-scan.  Yellows are heights above 765m, which is the 735m ASL ground level at TUR radar plus assumed 30m antenna height above ground.  (Relates to Figure 3.)

---

## Referee Comment (RC4)

Review of "Monitoring and quantifying wind turbine clutter in DWD weather radar measurements"
Author(s): Michael Frech et al.
(michael.frech@dwd.de)
MS No.: egusphere-2025-4957
Reviewer:  Norman Donaldson,  Environment and Climate Change Canada (Ret'd)

**Overall Quality**

The overall quality is very good.  I see no serious issues with methodology and the methodology is novel enough to deserve publication.

The work itself documents in some detail how wind farms impact radar measurements, both traditional (reflectivity and Doppler velocity) and dual polarimetric, which is important for other weather radar users.

My only qualification to that is the section on blockage, which I would describe as "indicative" rather than conclusive.   Trying to assess blockage by wind farms is exceedingly difficult. This is possibly the best attempt I have seen, even if not conclusive, so it should be presented.

I recommend publication after the authors consider my comments below.  I would characterize the changes as minor.

**Content Comments**

Radars :  Please give more description of the TUR and UMM radars:  latitude ,longitude, height of antenna ('horn height"), resolution of radar bins (1° x 250m?).  Frech (2017) gives other specifics like antenna size and wavelength but that could be repeated.  Possibly a small table.

Line 4:  "There are currently no filter methods that can reliably separate wind turbine clutter from desired weather information."  This is true for operational radar sampling, but I think there are techniques that do work with IQ data using a very large number of samples.

Line 15 "traditional radar reflectivity."  What does the word "traditional" mean here?

Line 16:  Maybe start new paragraph for the blockage discussion?

Line 94:  You point out that NCP does not isolate WTC from other static clutter sources so it is unsuitable by itself.   Operationally, we want to identify all clutter. Why is there a focus on

only finding wind turbines specifically?  Is that for research/regulatory purposes or is there an operational reason to distinguish WTC from other clutter?

Line 100:  Turbines are fairly isolated in range.  Less so in azimuth.  The method seems to be using only range isolation.   This is subject to the further qualification that the tower/mast is very isolated in range, but if a turbine is facing normal to the radar radial, the blades extend up to 100m along the radial in each direction.  I see that the detection algorithm is reporting values in front of turbines in Figure 3 and 8.  Is that an artefact of the algorithm or is it real detection of blades when they pointing toward the radar?

Line 105  Later it seems that CR is used only for the H channel.  Specify that here?

Line  127.  It is not explicitly stated how the naselle elevation of 1.0° was calculated.  I assume the difference in terrain height (about 70m) is included.  A quick look suggests that the bottom of the masts could be hidden by intermediate terrain and forest.  By the "height of the mast" I assume it is meant the height of the rotor axis.  (Ie the total height of blade tip at its highest should be mast + naselle + blades).

Figure 3:  The colour bar does not correspond to the figure.  The colour bar shows pure grays, but the figure is using colours with a magenta/purple shade.   The figure is deceptive because it implies the radar antenna is higher than all surrounding terrain.  Google Earth says the ground height is about 800m at turbines TUR2 and TUR3, and the height at the radar is about 735m.  That means the antenna would need to be at 65m above ground to equal the ground height at the turbines.  Maybe the radar is that tall?!  <See comment about providing details of the radar siting.>

Regarding Figure 3:  Maybe add  "The proposed detection method has highlighted two wind turbines at 12.4 km, 56.8° which were missing from our turbine database."    At coordinates (48.646°, 9.924°) there is a pair of turbines visible in Google Earth.  The figure has no X's there.  **<<After writing this I saw the discussion around Line 172.  Move that to here?>>**

Figure 3:  Is it my imagination or are there different intensities of yellow used for the detections?  At farms where I would expect to be the worst WTC the colours seem to be brighter yellow.

Figure 3:  It is not stated what elevation angle this data is from.  Terrain following?

Line 145:  Mention in main text that this from only one of the two turbines?

Figure 4  Change "wea" in title?   Is it easy to replace the WIGOS id with a name.  FYI  I could not find this ID in the OSCAR database, but I have little experience with WIGOS, so it might be somewhere else.

Figure 8:  As with Figure 3, colour bar does not correspond to the figure itself.

Figure 8 and Line 195  A better example of bad coordinates from the state database might be the echoes at  127°, 8.6km

Figure 8:  crosses are red not black.

Figure 8:  Comment: radar elevation angle of ground height using standard propagation might be better than simple height …. if it easily created.  (Same for Fig 3.)

Figure 10   In my opinion the upper limit on reflectivity colours should be higher.  I was surprised that the turbines seemed to be spread uniformly across 3°, until I realized that the observed values were far above the colour limits; colours were saturated so no detail below 3° in azimuth.   I suspect the reported reflectivities are in excess of 50 or 60 dBZ.  (This links back to the remarks around line 45 but isn't really a topic for discussion in the paper.)

Figure 11  What is the meaning of white?   I can guess that clutter has exceeded the ability to correct reflectivity,  but please state meaning.   The same comment for Figure 12

Line 235  "At 3.5° elevation, the wind farm has a larger effect on the spectral width."   Larger than what?  At first I though this mean larger than 1.5deg, but I assume it means larger than VRADH. (Regard Fig 11)

Figure 12:  Maybe comment on the red areas within the wind farm.  The QC has not caught these area.

Figure 14:  What is "Wea" and "NoWea" on the figures?   Maybe "WF" (Wind Farm) was intended, since discussion in the text indicates that the "Wea"  distributions are from the wind farm sector.  (Elsewhere WEA is WTC?)

Blockage section:  This is probably the best attempt to quantify blockage by a wind farm that I have ever seen.  However, I still think it is indicative rather than conclusive.  There seems to be a lot of variability/noise in the distribution.  Doing a good statistical estimate of the uncertainty in the estimates is not easy.   The only thing I'd suggest is trying to break the dataset in two (say by year) and comparing results from the two subsets.   Another potential objection is the assumption about the difference between data at 3.5 km and 6.5km being due only to the wind farm.  Is there any possibility that surface targets have differentially contaminated the data?  For example, there is a forest at 6.5km in one sector but not the other.  One might worry that the hill under the wind farm has blocked some signal.  The reviewer is almost certain that the hill is not an issue, but this should be stated.  (The

reviewer had exactly that potential situation. A look at data before a wind farm installation shows same the partial signal reduction we thought the wind farm caused. 😟 ) I am not saying these things are real issues, but they could be, even if I suspect they are not. I think the blockage section should remain despite my concerns, but if the authors have any responses they should please add them.

It would be useful for context to give some information about the turbines in the blockage assessment (hub height, blade diameter, mast diameter). One might add that the hubs are quite close to the middle of a beam at an elevation of 1.5° (reviewer assumed 100m mast) while the tips are at an elevation of about 2.3° when vertical and thus outside the nominal size of a 0.9° beam at 3.5° elevation (reviewer assumed 100m mast with 106m blades).

Line 331 "The beam blockage results clearly suggest, that wind turbine development in the 5 km radius must be avoided". My interpretation of the result is a bit more conservative, so I might delete "clearly".
- No comma in this sentence.

**Technical glitches:**

Throughout: should "WEA" be "WTC" (German to English for wind turbine clutter) except maybe "WF" where an entire wind farm seems to be indicated.

Style: I would write text like "depolarization ratio DR" with commas, such as "depolarization ratio, DR," but I know there is not consensus on this.

In several places "rangebin" -> "range bin"

Line 10 "WT" is not separately defined although it can be inferred from "WTC".

Line 141 A list of variable abbreviations is given. These are not defined until later in the paper.

Figure 5: It would be more visually pleasing if the frame on right were the same size as the frame on left. (Unimportant.)

Line 253 "extend" => "extent"

Line 219 "DR" has been defined, but "UDR" appears without definition.

Line 266  "since we do exclude" -> "since we exclude"  (Using "do" is slightly aggressive, it suggests an emphatic response to someone who suggested that you did not exclude.)

Line 267  and several other places, "ZH" and "Zh"  are used but  TH would be more consistent.

Line 273  "(1.4 – 30.0)".  What does this mean?

Line 330 "WT disturbance of DR".  I would say "on" rather than "of".   Not important.

Line 343  The word "ombrometer" is rarely used, so I suggest "rain gauge".   I even suspect that a quarter of readers will not know the word.

Figure 10:  FYI there are four turbines in the north of the wind farm where there are turbine echoes but no crosses.  Look near (52.1306N, 11.1613E)\

[Figure]

Coarse view of topography around TUR radar; SRTM30 data (~90m) as B-scan. Yellows are heights above 765m, which 735m ASL ground level at TUR radar plus assumed 30m antenna height. (Relates to Figure 3.)

[Figure]

Cross section from radar TUR to turbine "TUR2". Top: topographic height ASL. Bottom: radar elevation angle of topography assuming 30m antenna height. In elevation image the flat lines extend from lowest elevation visible to next visible topography,

---

## Author Comment (AC1)

**RC1 Response to review:**

We thank the reviewer for his constructive comments, which we address point by in the following:

Remark/Question:

In the detection method, the three central coefficients (around zero Doppler velocity) in combination with NCP to compute CR (line 98). However, later on in the paper, a new product called clutter power (CCORH) is shown in Figs 4-6. What is the relationship between CCORH and the clutter power used in the detection method? Looking at the figures, it is easy for a reader to use CCORH as a substitute for clutter power used to determine CR, and it doesn't make sense how you can get the CR curve shown in the right panel of Fig.5 with the two curves shown in the left panel of Fig. 5.

Response:

Thank you very much for this comment and feedback.

- Concerning "CCORH and the clutter power used in the detection method": CCOR(H/V) stands for "clutter correction (value)" (dB) and represents the amount of power removed by the clutter-filter at each range gate (with: clutter corrected reflectivity (dBZ) = uncorrected reflectivity (dBZ) + "clutter correction value" (dB)). Here, the clutter-filter itself is not nescessarily restricted to the three central DFT components and therefore can not be used as a possible substitute.
- Concerning "(CCORH) is shown in Figs 4-6": The CCOR(H/V) shown in Figs 4-6 represents the "output moment" after "range aggregation" (for DWD radar systems a 250 m range resolution for the output moments is used, aggregation of ten "25 m range gates").
- Concerning: "it doesn't make sense how you can get the CR curve shown in the right panel of Fig.5 with the two curves shown in the left panel of Fig. 5.":
  To explain, why the curves in our opinion do make sense, two cases:
  1. very slowly moving WT:
     NCPH values around 30..35 dB => no or minimal enhanced noise floor
     CCOR(H/V) values around '-45..-40 dB => dominant fixed target (as expected, WT tower)
     clutter ratio (CR) values around 45 dB .. 50 dB, as the "difference" between the noise floor and the three central coefficients is large
  2. moving wind turbine:
     NCPH values around 60 dB => enhanced noise floor (due to the rotating blades)
     CCOR(H/V) values around '-20..-15 dB => less dominant, due to the enhanced noise floor
     clutter ratio (CR) values around 30 dB, as the "difference" between the enhanced noise floor and the three central coefficients is now smaller

Corresponding changes in the document:

- line 54, Introduction: introducing "in-phase and quadrature-phase (I\&Q)",
- line 90, 2.1 Algorithm: added a requirement for the algorithm to be more precise
- line 91, 2.1 - list item 1: added "(25 m)" to be more precise
- line 98, 2,1 - list item 2: added "for each (25 m) range gate" to be more precise
- line 109, 2.1 - list item 5: added: "(25 m)" to be more precise

Remark/Question:

The sentence "But the spread is large indicating the wind turbine clutter signal most likely co-exists with a strong static clutter signal from the WT tower, which is decreasing with increasing rotation speed (Figure 6)" needs further clarification. I do not see a decreasing trend in CCORH in Figure 6. Also, why does the large spread in CR indicates the co-existence of WTC signal with a strong static clutter signal?

Response:

You are correct. The wording is not precise. The static clutter part is only slightly decreasing between 2 and 6 rpm, an remains more or less constant from 6 rpm on. (Figure 5, left panel). At the same time there is a continuous increase in NCPH. We have corrected the sentence accordingly in the revised document

Remark/Question:

For the data set shown in Fig. 4, how often was weather signals overlapping with the WTC signals? It would be interesting to break down the detection performance for cases when there is no weather versus when weather is overlapping with the WTC.

Response:

We are working on this, as this is a question we have asked ourselves. It turns out that we have to re-process the database as the data from the existing database is not sufficient to make clear distinction between weather / no weather cases. We will report about the results in a follow-up note.

**RC1 Technical Corrections:**

1. Line 56, "rangebins" should be "range bins". Please check the remainder of the manuscript for similar corrections.

Response: Thank you very much. We have implemented the correction. Furthermore, we have also fixed this in line 142, 143, caption figure 4, 153, 162, 179, caption figure 8, 188, 221, 236, 241, 310 and 338.

2. Line 96, insert comma after Doppler-spectra.

Response: Thank you very much. We added the comma.

3. Line 134: I would reword the sentence as "A data set comprising of at least one year of the typical meteorological situations would provide a more reliable validation."

good suggestion! We have replaced the original sentence.

4. Line 142: CCORH, CCORV, TH, TV, URHOHV, and UDR needs to be defined here when they first appeared in the manuscript.

Response: Thank you very much for this comment. We added the appropriate definition and additionally defined "uncorrected" to be more precise here.

5. Line 143: "3 by 3 rangebin area" should be "3-by-3 range bin area".

Response: Thank you very much. We have changed the text accordingly.

6. Line 145: "ration" should be "ratio".

Response: Thank you very much. We have corrected the spelling of the word.

7. Line 156: "40% to 10%" should be "10% to 40%"

Response: Thank you very much. We have corrected the order.

8. Line 181: "WEA" need to be defined.

Response: Thank you very much. The abbreviation "WEA" stands for the German word "Windenergieanlage" (wind turbine) and is unintentionally used in two places in the text. We have corrected these two instances and replaced the abbreviation "WEA".

9. Caption of Figure 8: "Black crosses" should be "Red crosses", "data based" should be "database", "rangebins" should be "range bins".

Response: Thank you very much. Yes, the naming of the color was incorrect. We corrected that and additionally enhanced the caption to be more precise.

10. Line 191: "WEA detection algorithm" should be "WTC detection algorithm"?

Response: Thank you very much. We addressed this adjustment in point 8 of the technical corrections.

11. Line 209: insert comma after "In Figures 10 and 12".

Response: Thank you very much. We added the comma.

12. Caption of Figure 12: "TH" was used previously to refer to reflectivity factor in dBZ. Here "Zh" is used. Please choose one designation and stay consistent.

Paper (wea_frech_rev1.tex): changed accordingly

13. Line 267: ZH is used here for reflectivity.

Response: Thank you very much for Your comment. ZH is incorrectly used here. It should be TH. We changed the text accordingly and added "uncorrected" to be more precise here. We considered adding "uncorrected" to the other text passages in Chapter 4, but decided against it for the sake of readability.

14. Figure 14: "WEA" and "NoWEA" are used in the legends. I think being consistent and using "WTC" and "NoWTC" would be appropriate.

Resonse: here beamblockage  is quantified using measuring differences in range in TH for an area with WT compared to a reference area without WT.  So the appropriate terms would be rather "WT" "no WT", since wea stands in German vor "Windenergieanlagen". We will change this in the plot.

15. Line 201, add comma after "-30 dB"

Response: Thank you very much. We added the comma (line 301).

---

## Author Comment (AC3)

**RC4: 'Reply on RC3', Norman Donaldson, 10 Dec 2025**

**Reply:** We thank the reviewer for his very careful review of our manuscript! The comments and suggestion were considered and we think they improved the clarity or our paper.

Review of "Monitoring and quantifying wind turbine clutter in DWD weather radar measurements"
Author(s): Michael Frech et al. (michael.frech@dwd.de)
MS No.: egusphere-2025-4957
Reviewer: Norman Donaldson, Environment and Climate Change Canada (Ret'd)

**Overall Quality**

The overall quality is very good. I see no serious issues with methodology and the methodology is novel enough to deserve publication. The work itself documents in some detail how wind farms impact radar measurements, both traditional (reflectivity and Doppler velocity) and dual polarimetric, which is important for other weather radar users. My only qualification to that is the section on blockage, which I would describe as "indicative" rather than conclusive. Trying to assess blockage by wind farms is exceedingly difficult. This is possibly the best attempt I have seen, even if not conclusive, so it should be presented. I recommend publication after the authors consider my comments below. I would characterize the changes as minor.

**Content Comments**

- **Radars:** Please give more description of the TUR and UMM radars: latitude ,longitude, height of antenna ('horn height"), resolution of radar bins (1° x 250m?). Frech (2017) gives other specifics like antenna size and wavelength but that could be repeated. Possibly a small table.
    - **Response:** We have added two tables with information on the radar locations Türkheim and Ummendorf and the WT location TUR2 and TUR3.

- **Line 4:** "There are currently no filter methods that can reliably separate wind turbine clutter from desired weather information." This is true for operational radar sampling, but I think there are techniques that do work with IQ data using a very large number of samples.

- o **Response:** There is some research out there (e.g. S. Torres presented a possible filter approach at ERAD2024 and the AMS radar conference in 2023), but those are so far not usable for operational applications; the results look promising, but only have been shown for case studies; commercially available signal processors do not come with WT filters. We aim at the development of a filter within the WICLEAN project ( see conclusions). We added that there is no filter available "for operational weather radars"

- **Line 15:** "traditional radar reflectivity." What does the word "traditional" mean here?
  - o **Response:** We removed "traditional".

- **Line 16:** Maybe start new paragraph for the blockage discussion?
  - o **Response:** A new paragraph is included. This makes sense.

- **Line 94:** You point out that NCP does not isolate WTC from other static clutter sources so it is unsuitable by itself. Operationally, we want to identify all clutter. Why is there a focus on only finding wind turbines specifically? Is that for research/regulatory purposes or is there an operational reason to distinguish WTC from other clutter?
  - o **Response:** There is primarily an operational reason. A proper classification (and quantification) of the clutter type is an requirement from developers that are responsible for radar based products, but also from forecasters (helps them to better interpret radar images). Another reason comes from agreements we have with some wind parks: they are supposed to shut down operation, if severe weather is in the pipeline which we do not want to miss with our radars. So far this option hasn't been used but we should be able to verify the proper shutdown of a wind park in such situations.

- **Line 100:** Turbines are fairly isolated in range. Less so in azimuth. The method seems to be using only range isolation. This is subject to the further qualification that the tower/mast is very isolated in range, but if a turbine is facing normal to the radar radial, the blades extend up to 100m along the radial in each direction. I see that the detection algorithm is reporting values in front of turbines in Figure 3 and 8. Is that an artefact of the algorithm or is it real detection of blades when they pointing toward the radar?
  - o **Response:** Thank you for your comment. Yes, you are correct: the method uses only one "ray" at a time. The fact that you also see detections in front of the turbines is due to step 5 of the method described in chapter 2.1: "Each local maximum found in this way is further extended so that all (25 m) range gates up to the previous and thefollowing minimum are marked.". This "extension" works on the "25 m range gates" (oversampling) which afterwards are getting combined to the final output "radial range resolution" (e.g.: the "250 m final output range resolution" is composed of ten "25 m range gates"). There, one

marked "25 m range gate" is sufficient to mark the whole final output "250 m range gate". So, it is by design, to be on the safe side.

- **Line 105:** Later it seems that CR is used only for the H channel. Specify that here?
  - **Response:** Thank you for your comment. You are correct regarding the detection algorithm (Chapter 2.1). We changed the text to be more precisely. In general: The estimation of CR runs for both polarizations independently. Throughout the paper, we show primarily results from the horizontal channel, as the results from the vertical channel are very similar.

- **Line 127**: It is not explicitly stated how the naselle elevation of 1.0° was calculated. I assume the di□erence in terrain height (about 70m) is included. A quick look suggests that the bottom of the masts could be hidden by intermediate terrain and forest. By the "height of the mast" I assume it is meant the height of the rotor axis. (Ie the total height of blade tip at its highest should be mast + naselle + blades).
  - **Response:** Your interpretation is correct.

- **Figure 3:** The colour bar does not correspond to the figure. The colour bar shows pure grays, but the figure is using colours with a magenta/purple shade. The figure is deceptive because it implies the radar antenna is higher than all surrounding terrain. Google Earth says the ground height is about 800m at turbines TUR2 and TUR3, and the height at the radar is about 735m. That means the antenna would need to be at 65m above ground to equal the ground height at the turbines. Maybe the radar is that tall?! <See comment about providing details of the radar siting.>
  - **Response:** Thanks a lot for pointing this out!  You are correct: antenna center height is 767,62 m, the total height of those wind turbines is about 1022 m (with respect to sea level); the height of the topography is 788 m.  We accidentally limited the plotted  range between -200 and 0 m, we now plot terrain height in a range between -200 and 100 m.

- **Regarding Figure 3:** Maybe add "The proposed detection method has highlighted two wind turbines at 12.4 km, 56.8° which were missing from our turbine database." At coordinates (48.646°, 9.924°) there is a pair of turbines visible in Google Earth. The figure has no X's there. <<After writing this I saw the discussion around Line 172. Move that to here?>>
  - **Response:**  In figure 8  we plot in addition the wind turbine location based on the satellite product and discuss the problem with up-to-date and correct data from the federal states. The  two wind turbines you point out are actually detected in the satellite product. However the two new wind turbines we investigate are not. The reason for this is simple: the product we use here was generated before the wind turbines were built. We are currently working on establishing an operational service of updated wind turbine locations every half year.

- **Figure 3:** Is it my imagination or are there different intensities of yellow used for the detections? At farms where I would expect to be the worst WTC the colours seem to be
brighter yellow.
  - **Response:** There is no color scale for the wind turbine clutter (WTC) detections. Once we find a WTC in more than 50% of the cases the corresponding rangebin is colored in yellow. Figure 9 shows an example where the WTC persistence in a range between 0 and 1 is shown.

- **Figure 3:** It is not stated what elevation angle this data is from. Terrain following?
  - **Response:** This shows the persistent WTC at an elevation of 0.5°. We have added this information into the caption.

- **Line 145:** Mention in main text that this from only one of the two turbines?
  - **Response:** Added that the 2-distributions are derived from TUR3 (one of the two wind turbines). Thanks for the hint.

- **Figure 4** Change "wea" in title? Is it easy to replace the WIGOS id with a name. FYI I could not find this ID in the OSCAR database, but I have little experience with WIGOS, so it might be somewhere else.
  - **Response:** Thank you for your comment. Concerning the WIGOS-ID: Yes, you are correct. The WIGOS-ID was assigned internally (0-276 DE) and is (so far) not useful outside DWD. We removed it therefore.
  wea is now replaced by "WT" (wind turbine).

- Figure 8: As with Figure 3, colour bar does not correspond to the figure itself.
  - **Response**: See response concerning figure 3 above.

- **Figure 8** and **Line 195:** A better example of bad coordinates from the state database might be the echoes at 127°, 8.6km
  - **Response:** Could be taken as well. In general we want to make the point, that there are quality issues with wind turbine locations from the official data bases, so we are quite happy that there is this satellite based source which can provide consistent WT locations across federal states and countries.

- **Figure 8:** crosses are red not black.
  - Response: corrected

- **Figure 8**: Comment: radar elevation angle of ground height using standard propagation might be better than simple height …. if it easily created. (Same for Fig 3.)
    - **Response**: We are working on a follow-up paper (where we look into the correlation of WTC and WT operation data in more detail) where we consider you suggestion. For now we want to keep the plot as is.

- **Figure 10:** In my opinion the upper limit on reflectivity colours should be higher. I was surprised that the turbines seemed to be spread uniformly across 3°, until I realized that the observed values were far above the colour limits; colours were saturated so no detail below 3° in azimuth. I suspect the reported reflectivities are in excess of 50 or 60 dBZ. (This links back to the remarks around line 45 but isn't really a topic for discussion in the paper.)
    - **Response**: You are correct, the colours are saturated at the location of the wind turbines. Setting the range of the color bar to a range of -10dBZ to 60 dBZ (see plot below) shows this. The point that we want to convey with this figure is to show how the bins surrounding the wind farm are or aren't affected by the turbines, not so much the impact at the location of the turbines itself. Hence, we opted to use a narrower color bar range to focus on more subtle differences. Figure 13 actually contains data from the exact same time with an extended colorbar.

[Figure]

- **Figure 11:** What is the meaning of white? I can guess that clutter has exceeded the ability to correct reflectivity, but please state meaning. The same comment for Figure 12
    - **Response**: Those are rangebins that are threshold using an SQI threshold of 0.25. This is mentioned now in the caption.

- **Line 235:** "At 3.5° elevation, the wind farm has a larger e☐ect on the spectral width." Larger than what? At first I though this mean larger than 1.5deg, but I assume it means larger than VRADH. (Regard Fig 11)

- o **Response**: We have reformulated the sentence. The wind farm has an clear effect on the spectral width if you compare the corresponding rangebins with the surrounding range bins where no possible wind turbine effect is seen.

- **Figure 12**: Maybe comment on the red areas within the wind farm. The QC has not caught these area.
  - o **Response**: Good point. However, we prefer not to go into the details on the DWD radar data quality details here. The purpose of this section is to provide a qualitative impression of the impact of this wind farm leading over to an analysis of beam blockage.

- **Figure 14**: What is "Wea" and "NoWea" on the figures? Maybe "WF" (Wind Farm) was intended, since discussion in the text indicates that the "Wea" distributions are from the wind farm sector. (Elsewhere WEA is WTC?)
  - o **Response**: We have changed this in the figure.

- **Blockage section**:

  This is probably the best attempt to quantify blockage by a wind farm that I have ever seen. However, I still think it is indicative rather than conclusive. There seems to be a lot of variability/noise in the distribution. Doing a good statistical estimate of the uncertainty in the estimates is not easy. The only thing I'd suggest is trying to break the dataset in two (say by year) and comparing results from the two subsets. Another potential objection is the assumption about the di□erence between data at 3.5 km and 6.5km being due only to the wind farm. Is there any possibility that surface targets have di□erentially contaminated the data? For example, there is a forest at 6.5km in one sector but not the other. One might worry that the hill under the wind farm has blocked some signal. The reviewer is almost certain that the hill is not an issue, but this should be stated. (The reviewer had exactly that potential situation. A look at data before a wind farm installation shows same the partial signal reduction we thought the wind farm caused.) I am not saying these things are real issues, but they could be, even if I suspect they are not. I think the blockage section should remain despite my concerns, but if the authors have any responses they should please add them.

  It would be useful for context to give some information about the turbines in the blockage assessment (hub height, blade diameter, mast diameter). One might add that the hubs are quite close to the middle of a beam at an elevation of 1.5° (reviewer assumed 100m mast) while the tips are at an elevation of about 2.3° when vertical and thus outside the nominal size of a 0.9° beam at 3.5° elevation (reviewer assumed 100m mast with 106m blades).
  - o **Response**:
    - ▪ Added a comment about the turbines in the windpark towards the southwest of UMD: "The wind park consists of 57 wind turbines, with rather small turbines. The median nacelle height of these turbines is 70 meters, the median rotor diameter is 60 meters."

- Added a comment about the robustness of the results: "We tested the robustness of the results by computing the reflectivity differences for various subsets of the data, stratifying it for example by year or by radar reflectivity mean values. The absolute numbers of the results vary based on the selection of the data. However, in all cases, the reflectivity differences at 3.5° elevation were very similar for both sectors, and there was always a clear difference of at least 0.3 dB between the two sectors at 1.5° elevation."
- A panoramic view from radar Ummendorf towards the south shows the relatively flat terrain, with no significant differences in terrain height in the two sectors used for the beam blockage analysis.

[Figure]

- **Line 331:** "The beam blockage results clearly suggest, that wind turbine development in the 5 km radius must be avoided". My interpretation of the result is a bit more conservative, so I might delete "clearly". - No comma in this sentence.
  - **Response**: we removed the "clearly". There remains to be a strong statement ("must"), which we want to make here.

**Technical glitches:**

- **Throughout**: should "WEA" be "WTC" (German to English for wind turbine clutter) except maybe "WF" where an entire wind farm seems to be indicated.
  - **Response**: Thank you very much. The abbreviation "WEA" stands for the German word "Windenergieanlage" (wind turbine) and is unintentionally used in two places in the text. We have corrected these two instances and replaced the abbreviation "WEA" (RC1 had also pointed this out to us.).

- **Style**: I would write text like "depolarization ratio DR" with commas, such as "depolarization ratio, DR," but I know there is not consensus on this.
  - **Response**: Thank you for your comment. We changed lines 214, 308 and related line 212.

- **In several places** "rangebin" -> "range bin"
  - **Response**: Thank you very much. We have corrected that (RC1 had also pointed this out to us.).

- **Line 10** "WT" is not separately defined although it can be inferred from "WTC".
  - **Response**: Thank you for your comment. We introduced "WT" with "wind turbine (WT)" in the abstract on line 12 (RC1 had also pointed this out to us.).

- **Line 141** A list of variable abbreviations is given. These are not defined until later in the paper.
  - **Response**: Thank you very much for this comment. We added the appropriate definition and additionally defined "uncorrected" to be more precise here (RC1 had also pointed this out to us.).

- **Figure 5**: It would be more visually pleasing if the frame on right were the same size as the frame on left. (Unimportant.)
  - **Response**: we keep it as it is.

- **Line 253** "extend" => "extent"
  - **Response**: Thank you for your comment. We corrected that.

- **Line 219** "DR" has been defined, but "UDR" appears without definition.
  - **Response**: Thank you for your comment. We have corrected this in accordance with the comment "Line 141" above.

- **Line 266** "since we do exclude" -> "since we exclude" (Using "do" is slightly aggressive, it suggests an emphatic response to someone who suggested that you did not exclude.)
  - **Response**: Thank you for your comment. We adopted that.

- **Line 267** and several other places, "ZH" and "Zh" are used but TH would be more consistent.
  - **Response**: Thank you very much for Your comment. ZH is incorrectly used here. It should be TH. We changed the text accordingly and added "uncorrected" to be more precise here. We considered adding "uncorrected" to the other text passages in Chapter 4, but decided against it for the sake of readability. (RC1 had also pointed this out to us.)

- **Line 273** "(1.4 – 30.0)". What does this mean?
  - **Response**: Thank you very much for Your comment. This should be dates giving the beginning and the end of the warm season. We have corrected that.

- **Line 310** "WT disturbance of DR". I would say "on" rather than "of". Not important.
  - **Response**: Thank you for your comment. We adopted that.

- **Line 343** The word "ombrometer" is rarely used, so I suggest "rain gauge". I even suspect that a quarter of readers will not know the word.
  - **Response**: Thank you for your comment. We adopted that.

- **Figure 10**: FYI there are four turbines in the north of the wind farm where there are turbine echoes but no crosses. Look near (52.1306N, 11.1613E)\
  - **Response**: As noted in comment to figure 8 we have the problem with up-to-date and correct data from the federal states with respect to wind turbine location. The three wind turbines you point out are actually detected by the satellite product. We are currently working on establishing an operational service of updated wind turbine locations every half year. For now we keep the figure as it is as this is the official data based from the federal state. There the overall impact of wind turbines on the radar data can be nicely demonstrated and discussed with these plots.

---

## Author Comment (AC4)

**RC5 - General SUGGESTIONS/COMMENTS**

"Monitoring and quantifying wind turbine clutter in DWD weather radar measurements" by M. Frech et al.

We thank the reviewer for his helpful and constructive comments!

**Comment 1:** This work reports the impact of WT on weather radar measurements; personally, I find it unbalanced: I mean, a great/relevant emphasis on the spectral component and a little one on key polarimetric variables such as rhoHV, Zv and Zdr.

**Response**:

Thanks for your comment. To our knowledge the DWD is the only weather service who has a dynamic WT detection algorithm running operationally on an IQ-data level. That algorithm specifically has not been developed to properly identify and quantify the WT impact on dualpol data. That will be an important aspect to be investigated in the WICLEAN project (see summary in paper). Since we now have WT operation data available the goal was to assess the WT detection algorithm for the first time with this unique data source. For completeness, we now show RHOHV and ZDR in section 3.1.

**Comment 2:** As far as the quality is concerned, I find it good with one exception: the authors fail in introducing and describing the variables. All the variables should be described in an introductory part of Sec. 2, I think. Currently, NCP is found only "inside" the algorithm description (step 1). CR is "hidden" inside step 2. CCOR is not even described! (or am I missing it?)

**What is CCOR?** Is it the estimate of the Log-power of the non-moving clutter that you are going to remove from the Uncorrected reflectivity factor? Is it intrinsically negative, because you ADD it?

Adding up, I recommend to describe ALL the variables at the beginning of Sec. 2.1.

**Response:**

For the reading flow, we think it is better to introduce the variables when they are used for the first time. You are correct with CCOR. Thanks for pointing this out! We have added an explanatory section when CCOR is mentioned the first time.

**Comment 3:** Since the WT polarimetric signatures are as interesting as the spectral ones, I would invite the authors to show the statistics of rhoHV and Zdr, explicitly; not hidden in a combined form inside the Melnikov-Matrasov DR for STAR weather radars.

**Response**:

We add RHOHV abd ZDR panels in the section. Note that DR is more and more used in weather radar networks. DR is aktually computed in the signal processor from eq. 5 in the cited Rhyzhkov paper. It is NOT computed from ZDR and RHOHV (output variables on processed rangebin level). At DWD DR (rather UDR) is used to better discriminate between non-meteorological and meteorological echoes. the "U"DR denotes a processing stream in the signal processor, where no clutter filter is applied.

**Comment 4:** There is also another IMPORTANT change to be implemented: I ask the authors to please **separate (split)** the counts when rotor speed is exactly 0 from those when 0<rs<0.5 rpm. This choice could improve the information content of Figures 4, 5 and 6. If the sample size of the observation period were large enough (something that is hard to evaluate a priori), then a peak would appear at small rotor speed, where the variability of TH (and TV) is the largest (much larger than at the energy production rotor speed). However, in order to foresee whether such variability could emerge in the DWD weather radars, it is necessary to know some fundamental info regarding the scan program such as:

- The antenna rotation speed in degree per second of the sweeps with different angles of elevation.
- The corresponding PRF of such sweeps

I ask the authors to add such info in a technical subsection, which gives also info on the radar sites (lat, lon, altitude of the antenna feed, … )

Furthermore, I think it would be interesting to compare observations with rotor speed larger than, say 0.5 or 1 m/s, with the case of not moving (or very slowly moving) WT.

**Response:**

We have looked into the suggestion to make a finer resolution for the 2-D histogram within 0-1 rpm. With the existing sample size and setup no additional insight could be found.

Note: A rotor speed of exactly 0 rpm (over the "measurement interval" of the wind turbine) is a very rare case (e.g. for maintenance), as this means that the wind turbine is being actively braked. Commonly (to our knowledge), wind turbines are tumbling slowly when not producing energy (e.g. the rotor blades not in the wind).

Information on PRF, antenna rotation speed and pulse length are now included in the text.

**Comment 5:** As far as Sec. 4 is concerned, I appreciate the efforts to give an evaluation of beam shielding; on the other hand, I find it somehow preliminary, indicative; we are still far from conclusive results. Consequently, I would make it more concise.

**Response:**

We certainly consider the data shown here as examples (we say so explicitly in the text). But those examples highlight the requirement, to keep WT out of the 5 km range. The question is: What would be a "conclusive" result? Based on observations like this, there will be always large variability (as indicated by the histograms we show).

**Comment 6:** Finally, I am well aware that this report represents just a case study; however, something more statistically robust would be nice, like quantifying over longer time periods and/or over a larger region the extent of the effect on radar observables as a function of rotation speed and distance (AZ, ELEV) to the WT.

**Response:**

This will be addressed in more depth using data from the WICLEAN project. Furthermore, as indicated in the paper, we are just receiving WT operation data since spring/summer 2025. We will revisit this analysis with a much larger data set (ideally when having a multi year data set)

**RC5 - DETAILED SUGGESTIONS/COMMENTS**

**Figures 4, 5, 6**: please, add another column in the raster figure in order to **separate (split)** the counts when rotor speed is exactly 0 from those when rs > 0. In this new additional column just put only the count when rs is exactly zero (perfectly still WT). Maybe, at low rotor speed, you may have BINS with different resolution, e.g., 0<rs<0.2 rpm, then 0.2<rs<0.5 rpm, …

**Response:**

We have looked into this, see response on comment 4 above. We were not able to obtain conclusive results.

**Lines 75-89:** I appreciate very much this introductory part which explains the reasons behind the spectral WT detection algorithm developed by DWD by evaluating in real time IQ data. Most of these findings have been shown in the papers already cited in the introduction, among others. Maybe they can be cited again here together with others such as, by way of example,

Angulo, I., Grande, O., Jenn, D., Guerra, D., and de la Vega, D., 2015: Estimating reflectivity values from wind turbines for analyzing the potential impact on weather radar services, Atmos. Meas. Tech., 8, 2183–2193

Bredemeyer, J., Schubert, K., Werner, J., Schrader, T., and Mihalachi, M.: Comparison of principles for measuring the reflectivity values from wind turbines, 20th International Radar Symposium (IRS), 26–28 June 2019, Ulm, Germany, 1–10, https://doi.org/10.23919/IRS.2019.8768171, 2019

**Response:**

We cite now both papers. Thanks for pointing this out!

**Lines 99:** Instead of "It is obvious that weather situations with a large spectral width are not addressed here", I think

"As a consequence, weather situations with a large spectral width cannot be addressed here" would much more appropriate.

**Response:**

Thank You very much. We have changed this accordingly.

**Lines104-105.** Sorry, I cannot possibly agree. With a ~ 150 m range resolution (coherent demodulation, hence ~c tau / 2), I would say it could be visible in 6-12 oversampled gates, or even more…

**Response**

We are not sure what the reviewer wants to point out here. We clearly say that oversampling is applied with the 25 m raw-rangebin resolution. We now say "five or even more successive range gates". Ok?

**Line 107:** "It is more likely a strong fixed target", "which includes the case of a perfectly still WT", I would add (or something similar).

**Response:**

Thank you for your comment. We have enhanced Line 93 accordingly (text position differs slightly from your original suggestion).

**Lines 128-129:** please, Keep it simple! Use in both line an HPBW of ~ 1 deg, do not use 0.9 deg in one line, and then 1 deg in the other … Consequently, in line 127, say the width is ~ 166 m, instead of 150 …

**Response:**

Thank you for your comment. We see what you mean. The half power beam width of the antenna is 0.9°. The calculation underlying figure 2 uses 1°. To resolve this we have deleted "(here a beam width of 1° was used)" and reproduced the figure with 0.9°.

**Line 128:** Please, write Half Power Beam Width (you may use the acronym) and NOT simply beam width, which is ambiguous.

**Response:**

We have changed "antenna beam width" on line 128 with "half power beam width of the antenna".

**Page 6, Line 141-142:**

The variables listed here (CR, NCP CCOR URHO UDR) should be introduced at the beginning of Sec. 2 , (I presume H and V are related to the polarization states, are not they.

Is URHO, the raw rhoHV, I mean not corrected for Signal-to-Noise ratio?

Is DR the Melnikov-Matrasov depolarization ratio? (36th AMS Conf. on Radar Meteorology and then Ryzhkov et al., JAMC 2017). I see you are introducing DR at page 12 (too late; as stated, the definition should appear ath the beginning of Sec. 2). And what is UDR? Does it mean using raw ZDR, I mean, using observed ZDR without any attempt to correct for attenuation? Or using URHO? Or what? In figure 10, bottom pictures you show UDR at ELEV 1.5 deg and 3.5 deg.

**Response:**

Thank You for Your comment. We enhanced the text accordingly (RC1 had already pointed this out to us.).

**Minor:**

I would avoid using the abbreviation "elevation" to mean "angle of elevation". You may use "EL" or "ELEV" or another abbreviation that you may have declared at the beginning of the manuscript.

**Response:**

Thank you for your comment. We enhanced the text accordingly (changed "elevation" to "antenna elevation" throughout the text).